# A Touchscreen-Based, Multiple-Choice Approach to Cognitive Enrichment of Captive Rhesus Macaques (*Macaca mulatta*)

**DOI:** 10.3390/ani13172702

**Published:** 2023-08-24

**Authors:** Antonino Calapai, Dana Pfefferle, Lauren C. Cassidy, Anahita Nazari, Pinar Yurt, Ralf R. Brockhausen, Stefan Treue

**Affiliations:** 1Cognitive Neuroscience Laboratory, German Primate Center, 37077 Goettingen, Germany; 2Leibniz-Science Campus Primate Cognition, 37077 Goettingen, Germany; 3Population and Behavioral Health Services, California National Primate Research Center, University of California, Davis, CA 95817, USA; 4Faculty for Biology and Psychology, Goettingen University, 37073 Goettingen, Germany

**Keywords:** cognitive enrichment, macaque, welfare, 3Rs, touchscreen, multiple-choice, preference

## Abstract

**Simple Summary:**

Across the last decades, animal welfare science has established that regular access to sensory, motor, and cognitive stimulation significantly improves captive animal’s well-being. In primates, in particular, cognitive enrichment protocols have been crucial to alleviate boredom and, more generally, several symptoms of compromised wellbeing. Despite this, cognitive enrichment practices have not received the same level of attention as structural and social enrichment. Consequently, captive animals are usually given ample climbing or grooming opportunities but are less frequently provided with intellectual challenges. This is especially problematic for primates and other species with high cognitive abilities and demands. Following and in order to expand upon recent scientific and technological progress, we developed a multiple-choice interface for touchscreen devices tailored to rhesus macaques housed at the facility of the Cognitive Neuroscience Laboratory of the German Primate Center. The interface allows the animals to flexibly choose between three tasks on a trial-by-trial basis, allowing them to switch activities as desired. Generally, our animals showed consistent task preferences, across time of day and weekly sessions, while also displaying proficiency in doing so. We believe that with a multiple-choice approach, it is possible to increase animal wellbeing by providing captive animals more opportunities to control their own environment, simultaneously providing researchers with a reliable and scalable method for cognitive assessment and animal training.

**Abstract:**

Research on the psychological and physiological well-being of captive animals has focused on investigating different types of social and structural enrichment. Consequently, cognitive enrichment has been understudied, despite the promising external validity, comparability, and applicability. As we aim to fill this gap, we developed an interactive, multiple-choice interface for cage-mounted touchscreen devices that rhesus monkeys (*Macaca mulatta*) can freely interact with, from within their home enclosure at the Cognitive Neuroscience Laboratory of the German Primate Center. The multiple-choice interface offers interchangeable activities that animals can choose and switch between. We found that all 16 captive rhesus macaques tested consistently engaged with the multiple-choice interface across 6 weekly sessions, with 11 of them exhibiting clear task preferences, and displaying proficiency in performing the selected tasks. Our approach does not require social separation or dietary restriction and is intended to increase animals’ sense of competence and agency by providing them with more control over their environment. Thanks to the high level of automation, our multiple-choice interface can be easily incorporated as a standard cognitive enrichment practice across different facilities and institutes working with captive animals, particularly non-human primates. We believe that the multiple-choice interface is a sustainable, scalable, and pragmatic protocol for enhancing cognitive well-being and animal welfare in captivity.

## 1. Introduction

Over the last three decades, considerable technological, experimental, and theoretical advancements have contributed to the refinement of cognitive assessment and enrichment of non-human primates (NHPs) in captivity. In fact, there is a substantial scientific literature where the fields of neuroscience, cognitive science, and experimental psychology intersect and from which autonomous, cage-based, computerized, testing and training protocols have emerged. Such protocols have the capacity to conduct cognitive assessments of and provide cognitive enrichment to NHPs simultaneously [1,2,3,4,5,6,7,8,9,10,11,12,13,14,15]. NHPs, who possess a rich set of cognitive abilities, often encounter only simple intellectual challenges in captivity. Incidentally, this mismatch between the level of skill possessed by the individuals and the level of challenge provided has been proposed as the main underlying factor for boredom [16,17]. In turn, boredom is associated with elevated levels of self-directed behaviors such as hair-plucking, stereotypies, and several other hallmarks of impaired well-being [18,19]. Computerized, cage-based, interactive systems can alleviate boredom by providing dynamic, autonomous, adaptable, and scalable cognitive enrichment [20]. Furthermore, these cage-based interaction systems have several other welfare- and scientific-related advantages for NHPs. (1) NHPs can interact with such devices from their home enclosure within auditory and olfactory (and sometimes even physical) contact with their group mates [3,4,5,15,21], and (2) they can do so at their own pace, often from within their home enclosure, with either limited or no physical constraint [22]. Such spontaneous, self-regulated engagement has been suggested to enhance the animal’s sense of agency and competence, which results in improved psychological well-being [1,9,23,24]. (3) A wide range of paradigms can be applied and improve data quality, whereby issues such as test-retest reliability (i.e., inconsistent results across multiple repetitions) and construct validity (i.e., appropriateness of the assessment with respect to the construct of interest) can be remediated (see [25,26] for comprehensive description on general statistical concerns regarding traditional cognitive testing in NHPs). (4) Testing and training protocols can be quickly evaluated for efficacy and fine-tuned if needed. (5) The advent of animal identification [3,5,27] allows for automated adjustments of cognitive assessment and/or enrichment protocols with respect to changes in the individual’s level of engagement and proficiency. Therefore, such flexible protocols can facilitate highly individualized autonomous training [5,28], as well as cognitive assessment and enrichment.

Accumulating evidence supports the idea that animals, including NHPs, are able and/or prefer to have the opportunity to choose between tasks [4,29,30,31,32,33,34,35,36,37,38,39]. There is indeed ample evidence that being able to control and choose how to engage enhances the sense of agency and competence, an aspect often lacking in captive environments. For example, Perdue and colleagues (2014) presented capuchin monkeys (*Cebus imitator*) and rhesus macaques (*Macaca mulatta*) with the option of choosing the order in which they would subsequently conduct a series of tasks (i.e., free choice) or conduct the same tasks in a predetermined order (i.e., forced choice). Subsequently, two sets of predetermined orders were created: a randomized order and an order mimicking the animal’s previous order selection. While reward probability and task demands were independent of the animals’ choice, all animals tested (six capuchin and five rhesus monkeys) preferred the free choice over the forced choice option, albeit there was high inter-individual variability. The authors suggested that the NHPs preferred to choose because having and making a choice is an act of control over one’s environment. This interpretation is in line with existing literature that links control of one’s environment with psychological well-being [34,40,41,42,43]. Beyond welfare-related benefits, choice paradigms with non-human primates have been demonstrated to be experimentally advantageous, allowing, for example, to assess the chromatic preferences in rhesus macaques [31] and food preferences in apes and monkeys [32]; as well as uncovering contrafreeloading behavior in Japanese macaques, a behavior in which animals “work to obtain food even though identical food is freely available” [33].

Building on the literature describing the beneficial effect of choice availability on animal welfare, we argue that integrating choice into cage-based, computerized protocols aligns cognitive enrichment and cognitive assessment even further. To this end, we designed a multiple-choice interface (MCI) and tested captive rhesus macaques with our autonomous, cage-based, touchscreen system eXperimental Behavioral Instrument [4,28]. Using the MCI interface, animals could select between three tasks to perform on a trial-by-trial basis: a reach task in which a fixed circle needed to be touched (*static reach*); a dynamic version of the reach task in which the circle bounced around the screen until touched (*dynamic reach*) and a task in which one picture was shown to the animal (from a pool of 126 pictures of NHPs) (*picture viewing*). While the first two tasks delivered a fluid reward upon a successful touch to the circle, we considered picture viewing to be rewarding on its own, and therefore, did not program this task to distribute a fluid reward. We quantified (1) the level of engagement and proficiency of 16 male rhesus macaques with an MCI protocol, and (2) the animals’ frequency of choice for each task depending on the position of the task stimulus on the MCI interface, time of the day, and over multiple sessions. Ultimately, by quantifying the animal’s level and style of engagement we assessed the feasibility of the MCI protocol as a cognitive enrichment tool. Animals operating the device purposefully and proficiently would suggest that a multiple-choice interface is indeed a suitable tool to provide cognitive enrichment and cognitive assessment for captive NHPs.

## 2. Materials and Methods

Research with non-human primates represents a small but indispensable component of neuroscience research. The scientists in this study are aware and are committed to the great responsibility they have in ensuring the best possible science with the least possible harm to the animals [44,45]. This study is part of our corresponding efforts [46,47,48].

### 2.1. Animals and Housing

The study was conducted on 16 adult male rhesus macaques (*Macaca mulatta*, 5–17 years of age, mean 10 years) belonging to seven social groups (two to four individuals per group, see Table 1), all already capable to perform various cognitive training paradigms. Data collection occurred on days in which the animals were not involved in their main experimental routine. All animals were housed in accordance with all applicable German and European regulations in the facility of the Cognitive Neuroscience Laboratory at the German Primate Center (DPZ) in Goettingen, Germany. The facility provides the animals with an enriched environment, including a multitude of toys and wooden structures, access to outdoor and indoor space and associated natural as well as artificial light, and exceeding the size requirements of the European regulations. The light cycle in the facility is automatically controlled to achieve daily 12 h dark/light cycles (from 7:00 to 19:00). During data collection, animals had free access to water, and monkey chow, and were provided with fresh food (e.g., fruits, vegetables, nuts) between 12:30 and 14:00. All animals participated in previous cognitive experiments using the same cage-mounted device described in this study.

### 2.2. Apparatus

The eXperimental Behavioral Instrument XBI [4,5,28,49] is mounted directly on the animals’ enclosure and can autonomously run cognitive experiments as well as various enrichment protocols (Figure 1). The device can operate stand-alone for several hours, requires little to no human supervision, needs minimal weekly maintenance, provides video monitoring and recording of each session, and is fully integrated into the DPZ’s local area network. The device is comprised of a centralized computational unit (MacBook Air MacOS Catalina, 10.15), a microcontroller (Teensy 3.5) to acquire touchscreen information and to handle the reward systems, a touchscreen (15 inches, 30.4 cm by 22.8 cm), two peristaltic pumps, and a custom-made mouthpiece (placed at 24 cm from the touchscreen) for reward delivery. In this study, pictures of the animals were taken at the beginning of every trial by a camera placed above the touchscreen to associate each trial outcome with the correct animal using a custom-made Matlab script (Matlab 2020b^©^, The MathWorks, Inc., Natick, MA, USA).

### 2.3. Experimental Paradigm

During each session, the XBIs ran an interactive, multiple-choice interface, developed as an extension of a proof-of-concept exploring multiple-choice behavior in one adult male rhesus macaque [4]. The MCI is comprised of multiple tasks from which the animal can freely choose amongst each trial throughout the session. Importantly, the type and nature of each task in the MCI can be tailored to address specific experimental questions or provide specific enrichment activities. In the current study, we investigated the feasibility of using a multiple-choice interface as cognitive enrichment by providing three tasks to the animals: two versions of a motor task that were rewarded with diluted juice for correct responses, and a picture viewing task that delivered no fluid reward. Each trial was initiated by the animal selecting one of three stimuli (from here on referred to as ‘buttons’) at the bottom of the screen (i.e., taskbar; see Figure 1). The buttons (5 degrees of visual angle in diameter) differed in shape and were arranged horizontally. Each button shape was associated with one task and triggered the start of a corresponding task if touched. Regardless of the task selected, a picture of the animal was taken after a button was touched and before the corresponding trial was loaded. These pictures were used for manual animal identification after the session was complete. In a pilot version of the experiment, in which two groups and four animals participated, the arrangement of the buttons was randomized in every trial. All other animals underwent a version of the experiment in which the icons’ arrangement was pseudorandomized every 60 min instead of on every trial. This modification was meant to expose the animals longer to each configuration to promote learning and disentangle task preferences from side biases (potentially due to handedness).

The MCI interface was programmed and run by the open-source software MWorks (http://mworks-project.org, version 0.10, 1 July 2023). The two motor tasks (*static reach* and *dynamic reach* tasks) were structured in the same way. Upon task selection and a trial initiation, the unselected buttons disappeared and a red circle (target) of variable size (5 to 10 degrees in diameter with consequent chance levels 7.33% to 30.24% respectively) appeared in a random location of the screen. Here animals were required to touch the target to correctly complete the trial (hit), which triggered acoustic feedback (‘ding’) and 0.37 mL of fluid reward. Touching anywhere outside the target and inside the gray background (excluding the task bar) resulted in a failure, which triggered different acoustic feedback (‘error’) but no fluid reward. If neither the target nor the background was touched within 5 s from target onset (timeout) the trial was considered ignored and aborted without acoustic feedback or fluid reward. Regardless of the trial outcome, the *Home Screen* and task bar appeared again after an inter-trial interval (randomized between 1.5 and 2.5 s) so that the animal could choose again. While in the *static reach* task (the circle, i.e., Button 1) the target was stationary, the target bounced around the screen at a variable speed (randomized between 10 and 30 degrees/second on a trial-by-trial basis) until the end of the trial during the *dynamic reach* task (the square, i.e., Button 2). If the picture viewing task was selected (the rhombus, i.e., Button 3), a random NHP picture (from a pool of 126) was presented in the middle of the touchscreen for 5 s. This task did not require any additional action by the animal and did not deliver any fluid reward. All pictures displayed a variety of NHP species performing a wide range of behaviors in naturalistic and semi-naturalistic environments at the DPZ or its field stations. All pictures were shown on a gray background, directly above the task bar (Figure 1).

### 2.4. Experimental Sessions

Experimental sessions occurred on Saturdays, when the animals’ main experimental routine did not occur and in parallel with standard weekend husbandry routines. Sessions were started between 8:00 and 9:30 and were stopped between 16:30 and 18:00 for an average of 8 h (minimum of 7 h and 39 min; maximum of 10 h and 2 min), for six consecutive Saturdays. All animals were fed between 12:30 and 14:00. Due to other experimental reasons, one group was tested over six sessions (from Saturday to Friday, with a break on Sunday). Most of the sessions occurred in parallel, with three to five XBI devices running autonomously and unsupervised simultaneously.

### 2.5. Manual Animal Identification for Data Curation

A picture from the front camera was taken every time a button was selected by an animal. Pictures were then used offline to identify which animal initiated the trial. This information was ultimately appended to the curated data frame so that data analysis could be conducted by animal identity. A custom-made Matlab app (Matlab 2020b©, The MathWorks, Inc., Natick, MA, USA) was programmed ad hoc for the offline labeling of each picture, at a speed of ~50 pictures per minute.

### 2.6. Data Analysis

Besides the analysis on task preference described below, all data were curated in Matlab (Matlab 2020b^©^, The MathWorks, Inc., Natick, MA, USA) and the analysis of performance, as well as visualization, was conducted in Python 3.9.

#### 2.6.1. General Engagement Analysis

To estimate the uniformity of engagement within each given session, across animals and sessions, we (1) normalized the time of each trial initiation to the time of each session end (Figure 2b) and (2) computed the median of each of the resulting distributions to (3) ultimately identify at which proportion of each session animals had performed half of the trials (Figure 2c, see also [5,49,50]). A distribution centered at 0.5 would indicate that animals performed as many trials before as after the session midpoint. The resulting distribution centers at 0.47 (Q1 = 0.14; Q3 = 0.76) which is significantly different from a hypothetical distribution of 0.5 (two-sided *t*-test; *p*-value = 0.0115, t = −2.58, N = 92), indicating that animals engaged more in the first half of the session compared to the second half, see Figure 1c.

#### 2.6.2. Task Preference Analysis

To investigate whether the animals exhibited a preference for one of the three tasks, we fit a Bayesian Generalized Linear Mixed Model (GLMM) with the family specified as ‘categorical’ (picture, *static task*, or *dynamic task*; note that the picture task was the reference category for the model) and a logit-link function. We included the position of the button (left, middle, or right), time of day, and session as predictors to account for potential hand preferences, variation in engagement throughout the day, and development of preferences over time, respectively. Additionally, we evaluated correlations between all variables (all Pearson correlation coefficients were below 0.5) and covariates were z-transformed to a mean of 0 and a standard deviation of 1 to provide more comparable estimates [51,52]. As random effects, we included animal identity, the group identity, and day into the model with all possible random slopes to keep type I error rates at the nominal level of 0.05 [52,53]. Data were analyzed with the ‘brms’ package (version 2.16.3 [54]) in R (version 4.1.2: R Core Team, 2021), which in turn makes use of ‘Stan’, a reference computational framework for fitting Bayesian models [54]. Each model was run using four Markov Chain Monte Carlo (MCMC) chains for 2500 iterations, including 1000 “warm-up” iterations for each chain. We checked the convergence diagnostics of the model and found no divergent transitions. We also found that all R-hat values were equal to 1.00, and that visual inspection of the plotted chains confirmed convergence. We used weakly informative priors to improve convergence, to avoid overfitting, and to regularize parameter estimates [55]. The prior for each intercept was a normal distribution with a mean of 0 and a standard deviation of 1. For the beta coefficients, we used a prior with a normal distribution with a mean of 0 and a standard deviation of 0.5. For the standard deviation of group-level (random) effects, we used a prior with an exponential distribution with scale parameter 1. Lastly, we used a LKJ Cholesky prior with scale parameter 2 for the correlations between random slopes.

As we were interested in individual differences in task preference, we examined the estimates and credible intervals of the random effects of animal identity in the Bayesian categorical GLMM (see Figure 3). We considered that there was evidence of individual task preference when at least one of the three tasks had an estimate and credible interval above those of at least one other task. Model estimates are reported as the mean of the posterior distribution with 95% credible intervals.

#### 2.6.3. Task Proficiency Analysis

To investigate animals’ performance in the *static* and *dynamic* tasks a series of Pearson correlation tests were run with the Python package *statsmodel* (version 0.13.0) between the average hit rate (adjusted to the change in chance level dependent on stimulus size) of each animal at each speed value. Across the stimulus sizes, 5, 6, 7, 8, 9, and 10 degrees of visual angles, chance levels were estimated to be 7.33%, 10.53%, 14.41%, 18.81%, 23.99%, 30.24% and interpreted as the likelihood of touching the stimulus by chance, based on the stimulus (hit) to background (failure) surface ratio. As the *picture viewing* task required no interaction from the animal and could not be terminated before 5 s had passed, throughout the data analysis we only quantified the number of times this task was selected, to assess animal choice behavior in the task preference analysis described below.

## 3. Results

### 3.1. General Engagement

We first evaluated whether rhesus macaques’ engagement remained stable across sessions or declined due to the fading out of novelty effects (as fluid/food control and social separation were not applied in this study). On average animals conducted 418 trials per session (Q1 = 239, Q3 = 806; Figure 2a), with no evidence of a significant decrease between the first and the last session (assessed through an animal-wise partial correlation that accounted for variations in session duration, Table 1). Within sessions, however, we observed a decrease in trial number (two-sided *t*-test; *p*-value = 0.0115, t = −2.58, N = 92), indicating that animals engaged more in the first half of the session compared to the second half (Figure 1c, see *Methods 2.6.1* as well as previous publications [5,49,50]).

**Figure 2 animals-13-02702-f002:**
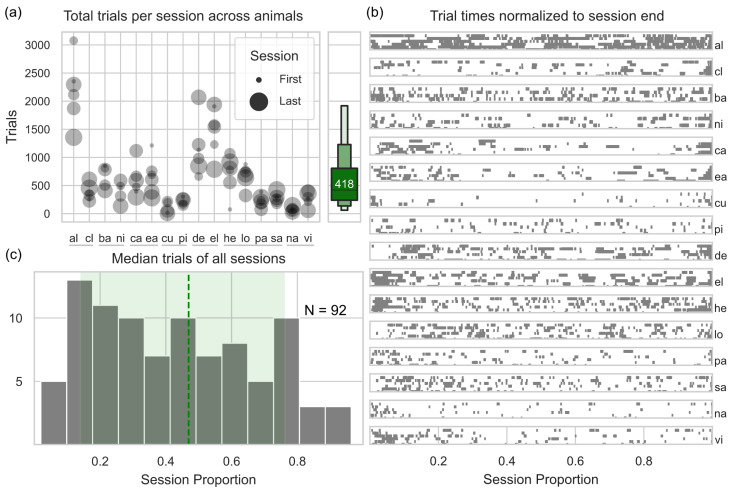
Level of engagement with the MCI. (**a**) On the left: number of trials of each animal at each session (dot size); on the right: median trials per session across all animals (green letter-value plot). Animals on the x axis are showed next to their respective partners, indicated by the solid black line highlighting animals belonging to the same group. (**b**) Distribution of all trial times (normalized to the end of each session, session proportion) for each animal and each session. (**c**) Distribution of all median trials times across all animals and sessions. Dashed green line and green shaded area represent the median of all median trials as well as the area between the median minimum and median maximum.

### 3.2. Task Preference

After establishing that the animals consistently interacted with the MCI, we assessed task preference across all animals and buttons’ positions, by quantifying which task was selected more often (see *Methods 2.6.2*). We found evidence that 12 out of 16 animals (75%) showed consistent task preference across button positions on the screen, time of day, and session. Of those indicating a preference, 10 animals (~83%) preferred the static task while two animals (animals *ni* and *sa*) preferred the *dynamic task*. More analysis results of the Bayesian model can be found in the Appendix A.

**Figure 3 animals-13-02702-f003:**
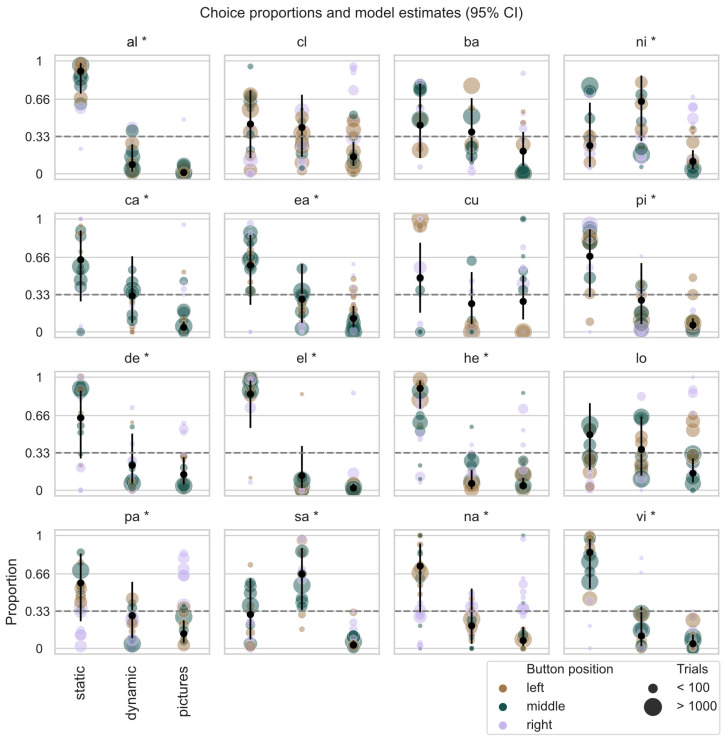
Task preference across all animals. Across all panels, each dot represents the proportion of trials in which the monkeys selected the three tasks, with the dots’ color representing the position of the button on the screen, for each given task and dot size reflecting the number of trials performed by each animal for each task and button position. Black dots and relative whiskers represent the Bayesian model probability estimates and the 95% credible intervals, respectively (calculated for the “middle” button position as we considered this position to be neutral with respect to potential differences in hand preference). The star symbol next the animal names indicate for which animal there is evidence of task preference.

### 3.3. Task Proficiency

Having established that most animals exhibited a task preference with the multiple-choice interface, we evaluated whether such engagement is purposeful or casual [18,56], and by extension, whether animals engage with the challenge using their own competence and potentially develop mastery. We found that the hit rate in the static task across all animals was already above chance at the smallest stimulus size we tested (namely 5 degrees of visual angle) and the hit rate increased with increasing stimulus size above chance level (Figure 4a). We used the adjusted hit rate (subtracting the chance level from the hit rate for each size independently) across all subsequent analyses to have a measure of performance that was not affected by relative changes in chance level due to stimulus size. We, therefore, evaluated modulations in adjusted hit rate caused by the different speed values we used (10 to 30 degrees of visual angle) and throughout consecutive sessions (Figure 4b) in the *dynamic task*. We found a significant modulation effect of speed on the adjusted hit rate for 8 animals out of 16, no modulation for three animals, and insufficient number of trials for the remaining five animals (see Table 1). Nonetheless, across all animals, medium to small stimuli elicited the highest adjusted hit rate, when paired with low speed in the *dynamic task* (Figure 4c). The optimal stimulus speed and size for our animals, from a psychophysical point of view, was a size of 5 degrees visual angle and a speed of 10 degrees of visual angle per second.

**Figure 4 animals-13-02702-f004:**
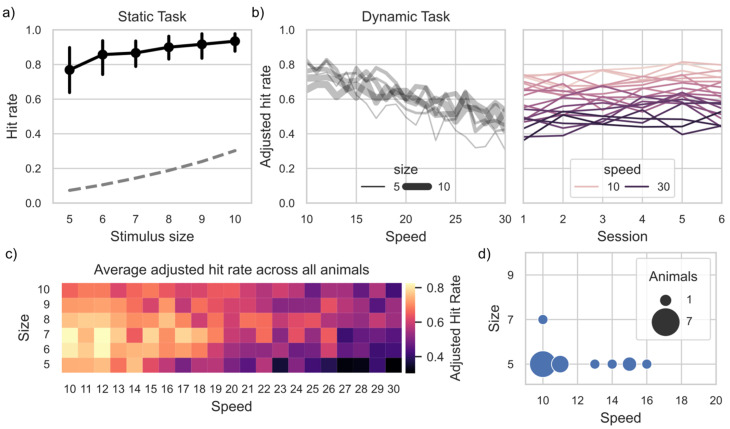
Proficiency in the static task (**a**) and dynamic task (**b**–**d**). (**a**) Absolute hit rate as a function of stimulus size in the static task across all animals (error bars are the percentile intervals at a 50% width, representing the range of half the distribution). Dashed line represents the theoretical chance level of the static task depending on the stimulus size, as the likelihood of touching the stimulus is function of the ration between stimulus and background size. (**b**) Hit rate adjusted to theoretical chance level (see panel a) as a function of stimulus speed and stimulus size (line thickness), on the left panel and as function of consecutive sessions and speeds (line thickness) on the right panel. (**c**) Heat map with average adjusted hit rate as function of both speed and size, across all animals. (**d**) Animal count of the size-speed combination with the highest adjusted hit rate.

**Table 1 animals-13-02702-t001:** On the left, descriptive statistics on animals’ general level of engagement (Figure 1a), with correlation values (Pearson), and respective *p*-values, on the trials per session across consecutive sessions. On the right, summary statistics of Pearson correlation between speed and hit rate (adjusted to size-dependent chance level, see methods). Overall, to account for multiple comparisons the adjusted alpha level is 0.003 (16 comparisons per analysis). *p*-values in bold mark significance reached for the respective animal.

	General Engagement (Figure 1a)							Correlation between Speed and Adjusted Hitrate (*Dynamic task* only Figure 4b)				
Group	Animal	Age	Avg. Trials per Session	Sessions	Pearson Corr.	*p*-Value (a = 0.003)	Total Trials	Pearson Corr.	Confidence Intervals (95%)	Number of Speed Values	Trials Performed	*p*-Value (a = 0.003)
1	al	13	2206	6	−0.65	0.22	13,068	−0.91	−0.97, −0.8	21	1622	**>0.001**
	cl	7	342	6	0.60	0.27	2229	−0.71	−0.87, −0.4	21	1179	0.0141
2	ba	17	785	5	−0.69	0.3	3364	−0.89	−0.95, −0.74	21	1025	**0.0003**
	ni	7	468	5	−0.38	0.61	1997	−0.53	−0.78, −0.12	21	426	0.0097
3	ca	12	532	6	0.14	0.8	3538	−0.27	−0.65, 0.22	18	46	**>0.001**
	ea	11	649	6	−0.93	0.01	3937	−0.67	−0.85, −0.33	21	630	0.0085
4	cu	16	50	5	−0.20	0.77	491	−0.56	−0.8, −0.17	21	963	0.2771
	pi	12	247	5	−0.32	0.69	1119	−0.66	−0.85, −0.32	21	379	0.5265
5	de	11	1059	6	−0.005	0.99	6922	−0.62	−0.83, −0.25	21	499	**0.0009**
	el	11	1561	6	−0.39	0.5	8992	−0.78	−0.91, −0.52	21	1169	**0.0011**
6	he	6	792	6	0.5	0.39	4203	−0.59	−0.82, −0.18	19	76	**0.0030**
	lo	5	717	6	−0.39	0.51	4057	−0.55	−0.79, −0.16	21	663	**>0.001**
	pa	9	228	6	−0.15	0.8	1380	−0.56	−0.8, −0.18	21	208	0.0077
	sa	9	283	6	0.49	0.4	1697	−0.15	−0.54, 0.3	21	148	**0.0001**
7	na	6	66	6	0.05	0.93	436	−0.74	−0.89, −0.46	21	984	0.0079
	vi	9	277	6	−0.37	0.53	1588	−0.13	−0.54, 0.33	20	130	0.5895

## 4. Discussion

The present study aimed to investigate the feasibility of using a touchscreen-based, multiple-choice interface (MCI) as a cognitive enrichment tool for captive rhesus macaques. We developed an interactive interface, optimized for touchscreen interaction, which allowed the animals to freely choose between three tasks: a *static reach* task, a *dynamic reach* task, and a *picture viewing* task. Our findings demonstrated that the rhesus macaques consistently engaged with the multiple-choice interface, exhibited task preferences, and displayed proficiency in performing the selected tasks.

### 4.1. Engagement and Sustainability of Interaction

One important aspect of cognitive enrichment is to provide animals with engaging activities that promote well-being. Our results showed that rhesus macaques maintained a sustained level of interaction with the multiple-choice interface across multiple sessions. Despite the absence of food/fluid control or social separation, the animals actively participated in the tasks, indicating their intrinsic motivation and interest in the paradigm. This finding is consistent with previous studies showing that non-human primates (NHP) can engage with computerized systems autonomously and for extended periods [5,18,27]. We argue that such propensity, together with the dynamic nature of touchscreen paradigms, could be leveraged to develop adaptive training and testing paradigms that consistently trigger animals’ curiosity.

The sustained engagement observed in our study also suggests that the novelty effect did not account for the animals’ interaction, as engagement remained stable throughout the sessions. Furthermore, our analysis revealed that animals tended to perform more trials in the first half of the session compared to the second half. This pattern suggests that the animals may have experienced some degree of habituation or reduced motivation over time within each session. It is worth noting that the decline in trials within a session could be influenced by factors such as satiety or fatigue, which might have affected the animals’ motivation to continue interacting with the tasks.

Finally, in contrast to the study conducted by Perdue and colleagues [35], in our experiment after each selection, at each trial, the selected button remained on the screen instead of disappearing. While this choice reduced the space on the screen available to display task-relevant stimuli (e.g., the red circle or the pictures), we decided on this approach to further help the animals to understand the button-to-task association. We believe that this was instrumental in alleviating the cognitive load for the animals, especially considering that buttons changed position on the bar every hour.

### 4.2. Task Preference and Individual Differences

The multiple-choice interface provides the animals with the opportunity to select and switch between different tasks at any moment. Interestingly, our analysis revealed that the animals exhibited individual preferences for specific tasks. The *static reach* task was the most preferred among the animals, followed by the *dynamic reach* task. Only two animals showed a preference for the *dynamic reach* task over the *static reach* task. These individual differences in task preference highlight the importance of considering the animals’ individual needs and preferences when designing cognitive enrichment programs. Tailoring enrichment activities to individual preferences can enhance engagement and maximize the benefits to the animals’ psychological well-being. The preference for the *static reach* task could be attributed to its high value-to-effort ratio. The animals might have found this task more rewarding and easier to perform compared to the *dynamic reach* task. The *dynamic reach* task involved a moving target, which required more precise timing and coordination. The animals’ preference for the *static reach* task aligns with previous studies showing that non-human primates often choose tasks that provide the highest rewards with minimal effort [35]. Interestingly, finding two animals preferring the higher effort task (namely the dynamic task) could be interpreted as evidence of contrafreeloading behavior [33], a behavior in which animals are willing to put effort to gain food that was otherwise freely available. As this study was not designed to directly probe contrafreeloading, we advise caution with such an interpretation, which we consider only anecdotal and requires further controlled experiments.

### 4.3. Proficiency and Mastery Development

While assessing the animals’ performance in the tasks, we found that our monkeys exhibited proficiency in both the *static task* and in the *dynamic task*. In the *static reach* task, animals achieved hit rates above chance level across different stimulus sizes, demonstrating accurate target selection. The animals’ proficiency in the *dynamic task* was only marginally influenced by the combination of stimulus speed and size and surprisingly, the medium to small stimuli paired with low speed elicited the highest adjusted hit rate. Albeit anecdotal, this finding might reflect a propensity for a flow-like state among our captive rhesus macaques. It appears that our animals found it optimal to engage (and perhaps even enjoy) when the task provided a moderate level of challenge, like the concept of *flow* experienced by humans during immersive activities. The animals might have reached higher levels of eye-hand coordination in the *dynamic task* in trials with a higher level of challenge. This observation aligns with the idea that animals, like humans, are motivated by the opportunity to achieve a state of optimal arousal and skill utilization [57].

### 4.4. Practical Implications and Future Directions

The touchscreen-based, multiple-choice approach presented in this study has several practical implications for providing cognitive enrichment to captive rhesus macaques. The multiple-choice interface, implemented in the eXperimental Behavioral Instrument (XBI), can be integrated as a standard cognitive enrichment practice in facilities and institutes working with captive animals, particularly non-human primates. The high level of automation and the potential to tailor the tasks to individual preferences (when real-time animal identification is available) make the approach scalable, sustainable, and pragmatic for enhancing cognitive well-being and animal welfare. Future research could explore further modifications and refinements to the multiple-choice interface and the tasks offered. The development of additional task options and the inclusion of more complex cognitive challenges could provide even greater enrichment opportunities for the animals. Moreover, investigating the long-term effects of the multiple-choice approach on the animals’ well-being, cognitive abilities, and social dynamics would provide valuable insights into the efficacy and potential benefits of this cognitive enrichment tool.

We believe that the time has come to consider cognitive enrichment practices as essential as the other more established types of environmental enrichment. Cognitive enrichment practices can consistently exercise the animal’s species-specific needs that have little opportunity to be expressed in captivity [6,17,24,58]. Consistently exercising cognitive capabilities, in turn, creates long-term benefits, such as: reduced distress and associated stereotypical behaviors; enhanced sense of competence, agency, and problem-solving in general and increased neuroplasticity against cognitive decline and impairment [20,34,59,60,61,62,63]. Moreover, with the type of cognitive enrichment here proposed we believe it will be possible to assess the role of genetic and environmental factors on cognitive development and decline [64,65,66], with crucial ethical implications for both humans and animals.

In conclusion, our study demonstrated the feasibility and effectiveness of a touchscreen-based, multiple-choice approach as a cognitive enrichment tool for captive rhesus macaques. The animals consistently engaged with the tasks, exhibited individual task preferences, and demonstrated proficiency in performing the selected tasks. Our approach has the potential to improve the psychological well-being of captive animals while providing researchers with a reliable and scalable method for conducting scientific research on animal cognition and welfare simultaneously [56,67].

## Figures and Tables

**Figure 1 animals-13-02702-f001:**
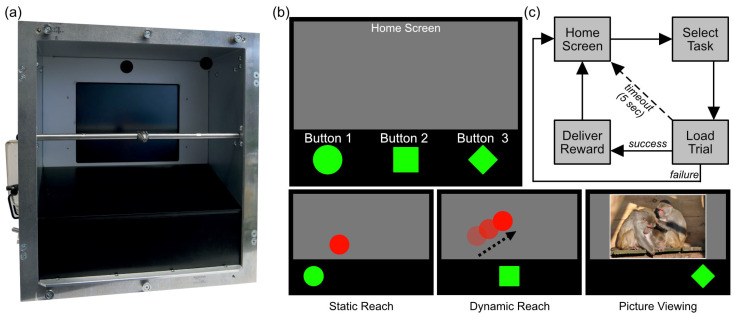
The eXperimental Behavioral Instrument XBI. (**a**) Front view of the XBI with the reward mouthpiece, the monitor and touchscreen, and the two cameras; (**b**) the multiple-choice interface (MCI) comprising the Home Screen, the Task Bar, and the Buttons leading to the three tasks; (**c**) flow chart of the transition between tasks and Home Screen depending on trial outcome.

## Data Availability

The data presented in this study are available on request from the corresponding author. The data are not publicly available due to being currently under further analysis.

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
