# Peer review of "A Touchscreen-Based, Multiple-Choice Approach to Cognitive Enrichment of Captive Rhesus Macaques (Macaca mulatta)"

_animals, 2023, doi:10.3390/ani13172702_

Round 1
Reviewer 1 Report
Calapai et al developed and tested a Multiple-Choice-Interface method for presenting cognitively enriching touchscreen tasks to non-human-primates. Seventeen macaques were each given 6 day-long touchscreen sessions in which they could decide on each trial which among three tasks they wanted to perform. Results from a GLMM indicated that a majority of the participants displayed a clear preference for the “Static Reach” task, and interestingly a couple of the participants displayed a preference for the more challenging “Dynamic reach” task. The method was clearly described and justified, the statistical model was well presented and executed. While the results were not altogether surprising-- the macaques overwhelmingly favored the easiest task that rewarded them with food—there was also some interesting nuances shown such as a minority of subjects favoring the more challenging task. This result suggests cognitive enrichment may be best practiced though individually tailored tasks and activities, and the MCI interface is a suitable way to achieve that end. Overall I believe the method and findings will contribute nicely to a growing body of research on cognitive enrichment and captive NHP stewardship, particularly with regard to providing opportunities for exerting choice and control as a means of enhancing overall welfare. I think the article merits acceptance in Animals after consideration of some minor points and suggestions for revision outlined below.
Minor Points
- Line 193: typo – “that” should be “of”
- Side-bias is always a concern for choices between multiple stimuli on a screen and so it was nice to see the authors attempted to mitigate it with the protocol described on lines 192-195. The authors should provide more detail about if and how side-bias was displayed by subjects, for example by calculating a reporting on a table a side-bias index value for each participating monkey. Behavioral adjustments following a shuffle of the icons would also be an interesting thing to report, e.g. how long after the icons were rearranged did it take for monkeys to resume clicking on their preferred task at the expected rate.
-As far as I am aware this study is the first to replicate and update the “menu” concept for NHP computer tasks since the SELECT paradigm used by Perdue et al. If I am interpreting Figure 1 correctly, a key difference seems to be that for the MCI the menu bar is omnipresent at the bottom of the screen with the current task’s icon appearing on it during a given trial, or all three choice icons appearing when the home screen is presented in-between trials. Since this seems to represent a new way of presenting multiple choices during the trial-flow, I think the authors should explain the rationale and perceived benefits of designing the user-interface in that manner compared to having the entirety of the screen be taken up by the task-at-hand without a menu-bar at the bottom of the screen.
-Since the authors are adding to a general body-of-knowledge about choice and preference in NHPs, there could be more background given on the history of preference testing methods. For example the pioneering methods of Humphrey (1971) for ascertaining color preference in Rhesus Macaques could be mentioned as well as more recent paradigms that used choices between photographic stimuli on touchscreens to assess food-preference in Japanese Macaques by Huskisson et al (2020). Also, a study by Ogura (2011) may be cited with regard to the method of reinforcing with stimuli rather than food, and the notion discussed that paper of “contrafreeloading” (animals preferring to work for their food) might also be a nice addition to the existing discussion on the possibility that NHPs may sometimes prefer more challenging tasks over easier ones (lines 409-412).
Humphrey, N. (1971). Colour and brightness preferences in monkeys. Nature, 229(5287), 615-617.
Huskisson, S. M., Jacobson, S. L., Egelkamp, C. L., Ross, S. R., & Hopper, L. M. (2020). Using a touchscreen paradigm to evaluate food preferences and response to novel photographic stimuli of food in three primate species (Gorilla gorilla gorilla, Pan troglodytes, and Macaca fuscata). International Journal of Primatology, 41(1), 5-23.
Ogura, T. (2011). Contrafreeloading and the value of control over visual stimuli in Japanese macaques (Macaca fuscata). Animal cognition, 14, 427-431.
Author Response
Dear Reviewer please find the point-to-point response attached.
Kind Regards
Antonino Calapai

Reviewer 2 Report
I commend the authors for such an interesting article regarding the application of touchscreen technologies as a cognitive enrichment for non-human primates. Particularly, in these species the enrichment provided needs to stimulate them physically and psychologically. The present findings show the future field of application of MCI. I left some minor comments regarding the manuscript.
As a general comment, please, revise the Instructions for Authors or the template of the journal to amend the in-text citation style (e.g., it must be numbered consecutively, inside brackets [1,2,3], or as Perdue et a. [20]). The reference list needs to be adjusted as well.
Simple summary. I recommend clearly stating what is the aim of the study (in the simple summary, abstract, and introduction). The simple summary needs to mention that the study was performed in rhesus macaques. It also needs to mention that the study was performed in a Neuroscience Laboratory. By what is described in the simple summary, it is not clear if the experimental phases were in a research center or in a zoological institution (conditions between both environments are quite different).
Abstract. In the sentence “To fill this gap, we developed an interactive, multiple-choice interface for cage-mounted touchscreen devices that animals can freely interact with.”, I recommend writing “rhesus monkeys (Macaca mulatta)” instead of just “animals”. Following my previous comment, the Abstract needs to give an idea about the study population (16 rhesus monkeys), the place where the study was performed (Neuroscience Laboratory), and how long was the evaluation period. Also, although the authors correctly state the results and that monkeys exhibited preferences, consistency, etc., there is no mention of what parameters did the authors assess in the interactive interface to reach these conclusions.
Line 40: Define the abbreviation “MCI”.
Lines 127-131. I’m not sure if these lines fit into the Methods section. Here I would expect to see an Ethical statement. Also, it is not clear what the authors mean by “(see also 7/31/2023 10:44:00 AM). Is this a reference?
Lines 133-134. What were the inclusion criteria to select these 16 adult male monkeys? Maybe the fact that they were already habituated to cognitive experiments, or that they were healthy animals. Please, include this information.
Table 1 seems to be more fitting in the Results section than Methods. Or were these parameters assessed in a pilot study to select the 16 monkeys? It is not clear why the Table is shown here when the reader hasn’t read anything about the sessions, correlation, trials, and so on.
Line 170. Correct the word “Behavioral”.
Lines 381-383. This is a very interesting finding that will be useful for future research because it is not always clear when a novel enrichment stops eliciting positive reactions from the animals. Knowing this information is important if the aim is to evaluate the effect of certain enrichment and how the repetition or level of engagement can alter the results. I suggest highlighting these findings by mentioning their application in future research with NHP.
Line 440. It would be interesting to add if the effect of cognitive enrichment depends on age or previous interaction of NHP with a similar type of enrichment, to determine if touchscreen-based enrichment can be provided to NHP regardless of their age (e.g., https://doi.org/10.1111/1365-2656.13857 or https://doi.org/10.1371/journal.pone.0109393).
Decision: Accepted with minor changes.
Author Response
Dear Reviewer please find our point-to-point response attached.
Kind Regards
Antonino Calapai
